# Dexmedetomidine Protects Cerebellar Neurons against Hyperoxia-Induced Oxidative Stress and Apoptosis in the Juvenile Rat

**DOI:** 10.3390/ijms24097804

**Published:** 2023-04-25

**Authors:** Robert Puls, Clarissa von Haefen, Christoph Bührer, Stefanie Endesfelder

**Affiliations:** 1Department of Neonatology, Charité—Universitätsmedizin Berlin, 13353 Berlin, Germany; robert.puls@charite.de (R.P.); christoph.buehrer@charite.de (C.B.); 2Department of Anesthesiology and Intensive Care Medicine, Charité—Universitätsmedizin Berlin, 13353 Berlin, Germany; clarissa.von-haefen@charite.de

**Keywords:** hyperoxia, oxidative stress, cerebellum, dexmedetomidine, postnatal developing brain, newborn rat

## Abstract

The risk of oxidative stress is unavoidable in preterm infants and increases the risk of neonatal morbidities. Premature infants often require sedation and analgesia, and the commonly used opioids and benzodiazepines are associated with adverse effects. Impairment of cerebellar functions during cognitive development could be a crucial factor in neurodevelopmental disorders of prematurity. Recent studies have focused on dexmedetomidine (DEX), which has been associated with potential neuroprotective properties and is used as an off-label application in neonatal units. Wistar rats (P6) were exposed to 80% hyperoxia for 24 h and received as pretreatment a single dose of DEX (5µg/kg, i.p.). Analyses in the immature rat cerebellum immediately after hyperoxia (P7) and after recovery to room air (P9, P11, and P14) included examinations for cell death and inflammatory and oxidative responses. Acute exposure to high oxygen concentrations caused a significant oxidative stress response, with a return to normal levels by P14. A marked reduction of hyperoxia-mediated damage was demonstrated after DEX pretreatment. DEX produced a much earlier recovery than in controls, confirming a neuroprotective effect of DEX on alterations elicited by oxygen stress on the developing cerebellum.

## 1. Introduction

Extremely and very preterm infants have a higher risk of neonatal morbidities [1]. The risk of oxidative stress is unavoidable in premature infants because, at the time of prematurity, they are still at a stage of development that necessarily requires a hypoxic environment. Thus, early exposure of the premature organism to room air and quite possibly intermittent hypoxic phases in combination with an immature antioxidant enzyme system inevitably generates oxidative stress [2]. The development of oxidative stress is based on different mechanisms, whereby oxidative stress caused by free radicals (reactive oxygen species, ROS) appears to be mainly responsible for several morbidities in premature infants [3,4,5]. The generation of ROS in the neonatal period is associated with oxygen supplementation, mechanical ventilation, and apnea-induced hypoxia in the context of respiratory instabilities in preterm infants [6,7]. A variety of studies assessing the short- and long-term effects of prematurity on the developing brain and immature lungs [2] have classified various diseases as “oxygen radical diseases in neonatology” [3,7,8].

Oxidative stress can cause lasting damage to biomolecules when the balance is disturbed by high levels of ROS and low antioxidant defenses, as evidenced in preterm infants [9,10]. The developing brain is highly vulnerable to damage from oxidative stress [11,12] which can lead to irreversible damage [6]. The last trimester of pregnancy is crucial and sensitive for a variety of neuronal network activities, such as synaptogenesis, neuronal migration, and myelination, which are finally important for neuronal plasticity [13]. In this phase, the immature organism is unable to counteract these stressful situations, so regulatory mechanisms are not fully activated, and this often leads to lasting changes in cellular neuronal processes, but also in cardiovascular, immunological, metabolic, and neuroendocrine responses [14].

In particular, the more finely tuned motor and cognitive functions and the higher rates of lifelong impairment in preterm birth [15,16] are more frequently the subjects of experimental and clinical research. The cerebellum, important for motor and cognitive functions [17], increases in volume remarkably in the third part of pregnancy and thus may be particularly vulnerable [18,19]. Impairments in these functions are then also associated with impaired psychological morbidities and socio-emotional behavioral disorders, such as attention deficit hyperactivity disorder (ADHD) and autism spectrum disorders (ASDs) [20,21,22,23,24,25,26].

Oxygen overload of preterm infants by birth per se as well as by required oxygen supplementation leads to oxidative stress, which has been proven to affect cerebellar functions as well. Interventions are directed towards adapted oxygen supply, which is currently not standardized, and antioxidative therapies, which reduce the harmful free radicals, since oxidative stress situations are unavoidable. Endogenous or exogenous antioxidants, such as enzymes, vitamins, and drugs, are able to neutralize the excess of free radicals and protect cells from the harmful effects of ROS [5,27,28]. Antioxidant strategies that reduce the pathological effects of oxidative stress, such as erythropoietin, vitamins A and E, nitric oxide, N-acetylcysteine, melatonin, and caffeine failed to show protective effects [28,29,30]. Consequently, a therapeutic approach would be conceivable that minimizes oxidative stress and neurodegeneration in extremely and very preterm infants with unavoidable exposure to room air and the equally unavoidable use of anesthetics, analgesics, and sedatives. 

Dexmedetomidine (DEX) seems to be a viable candidate as a highly selective α2-adrenoreceptor agonist with sedative, analgesic, and antianxiety effects. Experimental studies showed neuroprotective effects, clinically complemented by a reduced need for additional sedation and by the reduced use of opioids and benzodiazepines without suppressing respiration. Antioxidative mechanisms in the sense of a reduction of oxidative stress and subsequent anti-neurodegenerative and anti-inflammatory effects have already been demonstrated [31,32,33], as well as neuroprotection by DEX in the toxic oxygen model of our group [34,35,36].

Due to the paucity of available experimental studies on the effects of DEX with oxidative stress on the developing brain, this study examines the effect of DEX on the oxidative stress response after acute injury and subsequent recovery in a hyperoxia-mediated cerebellar injury model of the neonatal rat.

## 2. Results

To investigate the effects of oxygen toxicity with and without DEX on the developing cerebellum, 6-day-old rat pups were exposed to 80% hyperoxia for 24 h. Comparative analyses were performed in relation to the control animals that were kept under normoxia conditions. Furthermore, both oxygen-treated groups were divided into animals with and without DEX. Similarly, the animals were examined with different survival times for the acute and longer-term impairments, i.e., immediately after hyperoxia at P7 and after recovery on room air always in the presence of the providing mother at P9, P11, and P14.

### 2.1. Dexmedetomidine Exerts Anti-Apoptotic Effects following High Oxygen Exposure

To verify the effects on neurodegeneration in the cerebellum, cleaved CASP3-positive cells were evaluated, albeit without glial or neuronal classification. The induction of caspase-dependent cell death via caspase 3, an important performer of the apoptotic pathway, could be traced at P7 and P9 in the tissue sections under hyperoxic exposure (Figure 1) and was confirmed after quantification (Figure 2A). At the RNA transcript level, the increased *Casp3* at P7 and P9 was confirmed, and increased *Casp3* expression at P11 was now documented as well (Figure 2B). For data on caspase-independent cell death, the transcript level of mitochondrial *AIF* was examined and showed induction at P7 after hyperoxia (Figure 2C).

Pretreatment with DEX significantly reduced apoptotic cells directly after hyperoxia as well as at P9 (Figure 1 and Figure 2A). RNA-level analyses for *Casp3* (Figure 2B) supported this effect for P9 as well as for P11, in that expression was reduced by DEX to at least the control level, except for P7. *Casp3*-minimizing effects for DEX also existed at P14, although there was no detectable hyperoxia-induced increase (Figure 2B). DEX also reduced the oxygen-induced increase in mitochondrial *AIF* at P7 and decreased *Casp3* transcription at P11 (Figure 2C). Animals treated with DEX alone under normoxia exhibited increased apoptosis in tissue sections, as well as increased transcription at P11 (Figure 2A,B). The complete data are presented in Appendix A.

### 2.2. Dexmedetomidine Counteracts the Hyperoxia-Induced Oxidative Stress Response

Hyperoxia induced a significant increase in lipid peroxidation products (thiobarbituric acid reactive substances, TBARS) immediately after hyperoxia exposure at P7 and persisted with recovery at ambient air until P9 (Figure 3A). Hyperoxia induced a significant increase in GCLC transcripts, a rate-limiting enzyme for glutathione synthesis, at P7 and P11 compared with the control (Figure 3B). *Nrf2* transcripts, a transcription factor for various antioxidant enzymes and proteins such as GCLC, were adequately induced by hyperoxia at P7 and P9 (Figure 3C). DEX was able to reduce lipid peroxidation at both P7 and P9 (Figure 3A), and this occurred in correspondence with a reduction of *Nrf2* at P7, P9, and P11 (Figure 3C). DEX also tended to decrease the expression of *GCLC* at P7 and P11 when administered as a pretreatment to hyperoxia, but this effect was not significant (Figure 3B). Superoxide dismutase (SOD), an important endogenous antioxidant enzyme, requires metal cofactors for its activity and conditions different isoforms such as cytoplasmic Cu/Zn-SOD (SOD1), mitochondrial Mn-SOD (SOD2), and extracellular Cu/Zn-SOD (SOD3). All three SOD isoforms increase transcription by high oxygen concentrations at P7 (Figure 3D–F); *SOD2* increased at P9 and P11 (Figure 3E). DEX had a reducing effect on the transcription of *SOD1* and *SOD2* at P7 and/or P11 (Figure 3D,E) but not of *SOD3* at P7 (Figure 3F). Later on, the expression of *SOD1* and *SOD3* appeared to be depleted, resulting in reduced expression at P14 for *SOD1* and at P9 for *SOD3* (Figure 3F) with no further effects from treatment with DEX. DEX application from normoxic-maintained animals had few effects on the antioxidant response, such as a decrease at P7 for *SOD3* (Figure 3F). The complete data are presented in Appendix A.

### 2.3. Dexmedetomidine Promotes the Attenuation of the Hyperoxia-Induced Inflammatory Response

Oxidative stress also implies the induction of inflammatory processes. The inflammatory response to the hyperoxic insult presented at the protein level for TNFα (Figure 4A), but almost more impressively at the RNA level (Figure 4B) with an increase in TNFα gene expression at P7 and/or P9, i.e., immediately after hyperoxia and persisting until P9. The combination of an oxidative stress situation with DEX decreased TNFα expression (Figure 4A,B). As demonstrated at the protein and RNA levels, there was a very significant increase in TNFα at P11 in the hyperoxia-DEX-treated group (Figure 4A,B). DEX without high oxygen insult had no effect on TNFα. The effect on the equally early proinflammatory cytokine *IL1β* seems to be interesting; either hyperoxia or pretreatment with DEX and DEX alone reduced *IL1β* gene expression at P7 and/or at P9 and P14 (Figure 4C). The inducible NO synthase (*iNOS*) transcript was induced by hyperoxia at P7 and P9. DEX was able to reduce the expression compared to the hyperoxia group at P7, P9, and P14. The complete data are presented in Appendix A.

## 3. Discussion

In this study, hyperoxia-induced neurodegeneration in the cerebellum of the newborn rat as well as inflammatory and oxidative stress responses were attenuated by DEX, reinforcing a possible neuroprotective effect of DEX on oxygen stress-induced consequences on the developing brain. Oxidative stress markers showed that a single-day acute exposure during the vulnerable phase of cerebellar development to toxic oxygen concentrations caused a significant oxidative stress response, as well as inflammation and cell death and that these returned to normal levels after no more than 7 days (until P14) of reoxygenation. However, a comprehensive reduction of the effects of hyperoxia damage was demonstrated under DEX pretreatment directly at the early damage time points. DEX thus induced a substantially earlier recovery of comparable reoxygenation without potent neuroprotectants. 

Hyperoxia has consistently been shown in a wide variety of oxygen injury models [37] to result in a broad range of neurological sequelae that may include psychomotor impairment, impaired learning, and impaired spatial memory [38,39,40,41]. Concerning the human situation of preterm infants, oxygen toxicity is not preventable [42] and is being studied in a variety of hypothesis-driven appropriate in vivo and in vitro hyperoxia injury models [37,41,43]. Neurons are particularly susceptible to oxidative stress in this regard [44]. Any alteration in the hemostasis of the pro- and antioxidative balance can lead to cellular damage in the immature brain [45]. The most important finding of this study is that DEX both prevented the oxidative damage induced by acute hyperoxia already directly after 24 h of action under hyperoxic insult and was able to reduce the apoptotic and inflammatory responses to the oxidative insult that persisted beyond the recovery phase. Although per se cellular damage after recovery to room air ostensibly reveals no sequelae, this subtle damage from oxidative stress to cellular compartments may be responsible for the impaired brain functions that have been reported initially rather for extremely and very preterm infants [2,5,30].

The present study showed that thiobarbituric acid reactive substances (TBARS) as a marker of lipid peroxidation increased after hyperoxia and malondialdehyde (MDA) remained elevated even after a further 48 h of recovery in room air. Cerebellar TBARS levels did not normalize after acute hyperoxia until 96 h under normoxia, indicating a reduction in oxidative stress. The damaging effect of ROS in neurons is associated with their antioxidant activity. As an intracellular antioxidant, reduced glutathione (GSH) plays an important role. GSH directly scavenges the hydrogen peroxide produced during the catalysis of the superoxide radical by SOD. As shown in previous work [35], 24 h of hyperoxia led to reduced GSH, and increased GSSG resulted in a reduced GSH/GSSG ratio. This implies a loss of antioxidant capacity of the glutathione system and confirms increased oxidative stress. A decrease in antioxidant capacity in the developing brain after hyperoxia is not new [5,6,34,46]. In this context, the rate-limiting step of the cellular antioxidant GSH is catalyzed by glutamate cysteine ligase (GCL), which consists of a catalytic (GCLC) and a modifying (GCLM) subunit [47]. GCLC is a down-regulated target of the redox-sensitive transcription factor Nrf2 [47]. Hyperoxia resulted in both a significant induction of *GCLC* mRNA expression in the developing cerebellum and an increase of the transcription factor *Nrf2*, which supports the antioxidative stress response. *Nrf2* showed direct effects on pretreatment with DEX and significantly decreased the induction of *Nrf2* beyond the acute phase to P11. The recovery of antioxidant activity with a concomitant reduction in oxidative damage was more effectively suppressed with DEX, although it returned to normal after seven days even without DEX. Cytosolic copper-zinc superoxide dismutase (SOD1) and mitochondrial manganese superoxide dismutase (SOD2) keep the cellular concentration of superoxide radicals physiologically low. These key enzymes are responsible for the dismutation of superoxide radicals to oxygen and H_2_O_2_ [48]. In the cerebellum, *SOD* transcript isoforms tended to be induced but inhibited by DEX. In terms of concentration dynamics, *SOD1* increases rapidly after birth and peaks around the second postnatal week. Delayed and only tendential induction of SODs as an antioxidant response reflects the impaired imbalance of antioxidant enzymes in the immature cerebellum, which reaches relative stability only at the mature developmental stage [49,50]. Unbalanced ROS production has direct effects on lipids, proteins, and DNA, which consequently leads to damage to cell structures and causes changes in mitochondria [51]. Induction of iNOS in the brain occurs in astrocytes and microglia induced by proinflammatory cytokines and pathogens [52]. Once expressed, iNOS produces high and sustained levels of nitric oxide (NO) that may well degenerate neurons under certain conditions [53]. One possible mechanism would be the reaction of NO with superoxide to form the neuron-toxic peroxynitrite [54]. By oxidative reactions or radical-mediated mechanisms, peroxynitrite reacts with macromolecules such as lipids or DNA, which are known to then result in altered cellular cell signaling up to oxidative damage and cell death [54]. The results of our analysis are consistent with increased oxidative stress causing inflammation and cell death, possibly via the toxic effects of peroxynitrite.

It is already known that oxidative stress is a potent trigger of apoptosis, which then contributes greatly to neuronal cell death in hyperoxia in addition to physiological apoptosis. ROS intrinsically induce mitochondria-dependent apoptosis via caspase-independent mediators, such as AIF, or through the release of cytochrome c and the downstream activation of caspases. AIF is thereby induced by ROS released from mitochondria [55]. The results of this study showed that hyperoxia induced a significant increase in *AIF* and *Casp3* at the mRNA level, confirmed by the cellular increase in cleaved CASP3-positive and thus apoptotic cells in cerebellar white matter. In accordance with the increase in oxidative stress response, increased cell death was detected, and as well the protective effect of DEX was determined. 

Hyperoxia-induced apoptosis is also involved in inflammation during this process [51]. This oxidative stress-induced cell death is accompanied by increased levels of proinflammatory cytokines. The pro-inflammatory cytokines TNFα and IL1β are very potent and represent cytokines that can trigger a cascade of events and influence the release of other cytokines. Massive production of pro-inflammatory cytokines may be involved in diverse neuropathological conditions. Consistent with oxidative stress and cell death, the pro-inflammatory cytokine TNFα was induced. Neuroinflammation has been shown to be a possible mechanism of oxygen-induced toxicity in the rat brain [35]. This was demonstrated in our study by the increase in TNFα in the cerebellum, which is consistent with previous reports for the immature brain [35]. This oxidative stress-induced cell death is accompanied by increased levels of proinflammatory cytokines. Induction of iNOS in the brain occurs in astrocytes and microglia induced by proinflammatory cytokines and pathogens [52]. Once expressed, iNOS produces high and sustained levels of nitric oxide (NO) that may well degenerate neurons under certain conditions [53]. One possible mechanism would be the reaction of NO with superoxide to form the neuron-toxic peroxynitrite [54] and peroxynitrite reacts with macromolecules [54]. The results of our analysis are consistent with increased oxidative stress causing inflammation and cell death, possibly via the toxic effects of peroxynitrite. Consistent with oxidative stress and cell death, the anti-inflammatory response in this damage model was enhanced by DEX.

As also shown in the present study, during recovery on room air after acute hyperoxia, the balance between pro- and antioxidant responses readjusts and returns to normal. Nevertheless, a single pretreatment with DEX appears to have protected against the initial oxidative imbalance. Based on the available data, the neuroprotective effect of DEX allows for several possible mechanisms of action [56]. Apoptosis is usually a consequence of antecedent damage and, of course, is essential in normal organ development. Anti-apoptotic properties of DEX would be explainable because the signaling pathways of essential pro- and anti-apoptotic proteins, such as BCL2, ERK1/2, and NfκB, were modulated and this resulted in inhibition of Casp3 activation [57,58,59,60,61]. Regarding the NfκB signaling pathway, neuroinflammation presents as a complex process with many players involved, such as pro-inflammatory cytokines, chemokines, and inflammatory cells. Both damaged/injured cells and glial cells release inflammatory mediators. Glial cells repair neurons and assist in regenerative processes. DEX exerted inhibitory effects on over-activated astrocytes, inhibited TNFα and GFAP, and induced BDNF [62,63,64], among others, which means that DEX modulated neuroinflammatory mediators. The most important mediator of inflammation is oxidative stress and in particular ROS. This is controlled by the redox-sensitive transcription factor Nrf2, which regulates the antioxidant activities of cells. 

The pro-inflammatory cytokine IL1β is an essential player in neurodegeneration and a downstream target of Nrf2. In this context, any injury leads to rapid and drastic induction of gene expression, but also chronic neurodegenerative diseases often show an increase in IL1β expression [65]. Here, no oxidative stress response for *IL1β* could be detected. The transcript levels at P7 were significantly reduced, different from the previous study of similar animals in the cerebrum [35]. Blockade of IL1 induction or its receptors resulted in a reduced extent of injury in most cases [35,46]. Considering the effects of IL1 signaling in the healthy brain and under pathological conditions, differences can be identified [66]. In the physiological brain, the presence of glial activation and the associated expression of cytokines indicates the possibility of an effective response to injury. Both glial activation and neurogenesis are processes modulated by IL1 [67,68]. IL1β protein has been detected in cells with a neuronal phenotype in the Purkinje cell layer of the cerebellum and the granular neurons of the dentate gyrus in the hippocampus [69]. Adult IL1 receptor (IL1R) expression is primarily restricted to the hippocampus and Purkinje neurons of the cerebellum [70,71]. After brain injury, IL1β and IL1R are mainly induced in microglia and astrocytes and promote their activation [65]. In this regard, it appears that neuronal IL1β induces changes in neuronal excitability, while on astrocytes the effects of IL1β mediate neuronal survival [72]. That the IL1 system exerts a direct modulatory effect on cerebellar Purkinje cells was demonstrated by Motoki et al. [71] in an IL1R knockout mouse model. In the current study, IL1β was not induced under hyperoxia and also displayed decreased transcript levels both with or without DEX. A relationship between the proinflammatory cytokine IL1β and an influence on neuronal cell activity remains hypothetical.

It is worth mentioning that the demonstrated effects resemble autophagy mechanisms, which include anti-apoptotic properties, inhibition of the inflammatory response, clearance of damaged mitochondria, and reduction of oxidative stress, which is detectable through interaction with several associated genes [73]. When toxic noxious agents are involved, the identification of underlying mechanisms seems more targeted. This study demonstrated that under oxidative stress activation of the Nrf2-redox-sensitive signaling pathway would explain an antioxidant effect of DEX with Nrf2-downregulation. Similarly, there was an increased rate of apoptotic cerebellar cells, accompanied by increased transcripts for *Casp3* and *GCLC*, probably triggered by the single DEX application already 5 days ago. However, autophagy, an adaptive degradation process, has recently been shown to play a crucial role in the protective effects of DEX [73,74,75]. Relevant studies indicated that autophagy was modulated under DEX in the presence of organ injury and multiplicity, suggesting activation of autophagy under DEX rather than inhibition [73]. As altered autophagy was frequently found in neurodegenerative insults [76], DEX treatment was also shown to affect autophagy-related processes [77]. If autophagy-associated mediators are modulated, then it seems very likely that other associated proteins and signaling pathways could also be altered, e.g., ROS, Nrf2, and Casp3 [78,79,80,81]. A short-term hyperoxic insult may induce sustained ROS and isoprostanes (8-iso-PGF 2), a biomarker of lipid peroxidation, beyond the exposure time [82]. Peroxidation of lipids can also alter the membrane lipid organization [83], causing changes in fluidity and permeability, and additionally, damage to mitochondria induced by lipid peroxidation can lead to further ROS generation. The DEX effect opposing in our understanding of protective mechanisms, under both normoxia and hyperoxia, as demonstrated for TNFα, cleaved *Casp3* and *GCLC* at P11, 5 days after a single application or exposure, could be autophagy-associated effects in terms of a counter-regulatory effect. Autophagy may be another regulator in the protective effect of DEX, and its role should be further investigated.

This study shows that pretreatment of dexmedetomidine markedly reduces hyperoxia-induced upregulation of oxidative stress responses, proinflammatory cytokines, as well as mediators of cell death in the developing rat brain. These results reinforce the protective effects of dexmedetomidine on neuroinflammation and neurodegeneration through antioxidant effects. The targeted use of antioxidant drugs in the hyperoxia-damaged immature preterm brain can be exploited for therapeutic purposes, as initial changes that occur already through the activation of anti-inflammatory and anti-apoptotic signaling pathways can have far-reaching protective effects. Premature infants often require sedation and analgesia, and commonly used opioids and benzodiazepines are associated with adverse effects. Dexmedetomidine appears to be a promising alternative based on clinical and experimental data Further studies should seek a better understanding of the mechanisms in order to determine the safety of its applicability in the NICU.

## 4. Materials and Methods

### 4.1. Animal Welfare

Time-mated Wistar rats (Janvier Labs, Le Genest-Saint-Isle, France) were kept under standardized conditions with a constant 12-h/12-h light-dark cycle, equal room temperature, and unrestricted access to food and water. Two days before the planned date of birth, the rats were kept in individual cages. As previously described [34,35,36,84,85], newborn pups remain with a lactating mother for the entire duration of the experiment. The dams were allowed to birth naturally and were disturbed for the first time to randomize the pups on postnatal day (P)4. We performed our animal experiment in accordance with the authorization granted by the local animal welfare authorities (LAGeSo, Berlin, Germany, authorization number G-0145/13 and G-139/18), which complied with institutional guidelines and ARRIVE guidelines.

### 4.2. Oxygen Exposure and Drug Administration

The rat pups were randomized according to litter and sex. Rodent pups, together with the lactating mother, were placed in a hyperoxia chamber (OxyCycler BioSpherix, Lacona, NY, USA) on P6 and were exposed to a continuous flow of a gas mixture of 80% O_2_ and 5% N_2_ at 37 °C for 24 h, as described previously [34,35]. Fifteen minutes before the onset of oxygen exposure, rat pups received an intraperitoneal injection (i.p.) of weight-adapted DEX (5 µg/kg; DEX; dexdor^®^, Orion Pharma, Espoo, Finland; dissolved in phosphate-buffered saline (PBS)) or vehicle (0.9% saline). Age- and sex-matched littermates remained as normoxic controls and received adequate DEX or vehicle (i.p.). The neonatal rats were separated into 4 treatment groups, namely a normoxia group (NO) with 21% O_2_/PBS (i.p.), a hyperoxia group (HY) with 80% O_2_/PBS (i.p.), a DEX-treated normoxia group (NOD) with 21% O_2_/5 µg/kg DEX, and a DEX-treated hyperoxia group (HYD) with 80% O_2_/5 µg/kg DEX. The levels of apoptosis, oxidative stress, and inflammation in the immature cerebellum were evaluated directly after hypoxia at P7 and after recovery in room air at P9, P11, and P14.

### 4.3. Tissue Preparation

All animals were anesthetized with an i.p. injection of ketamine (100 mg/kg), xylazine (20 mg/kg), and acepromazine (3 mg/kg) before being transcardially perfused with a phosphate buffer and phosphate-buffered 4% paraformaldehyde (PFA) as previously described [34,35,84,85]. The pups were analyzed at P7, P9, P11, and P14. After decapitation and opening of the skull, the complete cerebellar tissue was removed and immersed in 4% PFA overnight. Then, cryopreservation in liquid nitrogen and storage at ™80 °C was performed for the tissues after transcardial perfusion with PBS (pH7.4) preparing for gene expression studies. The post-fixed samples were subsequently prepared for dehydration with ascending ethanol series followed by embedding in paraffin.

### 4.4. RNA Extraction and Quantitative Real-Time PCR

Tissue procurement has already been described [84,86]. Briefly, the total RNAs from a half snap-frozen cerebellum were extracted using peqGOLD RNAPure™ (PEQLAB Biotechnologie, Erlangen, Germany), according to the manufacturer’s instructions. After DNase treatment, 2 µg of RNA was reverse transcribed into cDNA. 

According to the manufacturer’s instructions, cDNAs were amplified using the qPCR BIO Mix Hi-ROX (NIPPON Genetics Europe, Düren, Germany) and analyzed with the StepOnePlus real-time PCR system (Applied Biosystems, Carlsbad, CA, USA). Hypoxanthine-guanine phosphoribosyl-transferase (*HPRT*) was used to normalize gene expression, and the relative gene expression values were calculated using the 2^−∆∆Ct^ method [87]. PCR amplifications were carried out in triplets under the following conditions: 50 °C for 2 min, 94 °C for 2 min, 40 cycles at 94 °C for 5 s, and 62 °C for 25 s in an 11 µL reaction mixture consisting of 5 μL master mix, 2.5 μL 1.25 μM of each oligonucleotide primer, 0.5 μL 5 μM of the probe and 3 μL cDNA template (17 ng). The primer and probes for quantitative PCR were synthesized by (BioTez Berlin Buch GmbH, Berlin, Germany) and presented in Table 1.

### 4.5. Immunohistochemistry

As previously described [84,86]. After dehydration, cerebellums were paraffin-embedded, cut into 5 μm sections, and mounted onto SuperFrost Plus coated slides (Menzel, Braunschweig, Germany). Then, sections were deparaffinized in Roti-Histol (Carl Roth, Karlsruhe, Germany) and rehydrated in graded ethanol concentrations. Antigen retrieval was performed by boiling sections in citrate buffer (pH 6.0) in a microwave oven for 10 min at 600 W. After washing the sections with PBS, they were pretreated with 2N HCl for 30 min, washed with PBS three times and then incubated with 0.1% TX-100 and 0.1% TW-20 in PBS for 15 min. The primary antibody was monoclonal rabbit anti-cleaved caspase-3 (1:200 in antibody diluent (Zymed Laboratories, San Francisco, CA, USA), ASP175, (Cell Signaling Technology, Beverly, CA, USA), and the sections were incubated 48 h at 4 °C. After three times washing in PBS, the sections were incubated for 4 h at room temperature with secondary Alexa Fluor 594-conjugated donkey-anti-rabbit IgG (1:200 in antibody diluent, Thermo Fisher Scientific, Waltham, MA, USA). All sections were washed once in PBS. Nuclei were stained with 4′,6-diamidino-2-phenylindole (DAPI, Sigma, St. Louis, MO, USA) diluted 1:2000 in PBS (10 min, RT). After three final washes with PBS, sections were cover-slipped (Shandon Immu-Mount, Thermo Fisher Scientific).

The digitalization for cleaved caspase 3-positive cells was performed using a Keyence BZ 9000 compact fluorescence microscope with BZ-II Viewer software and BZ-II Analyzer software (Keyence, Osaka, Japan). For each time point, the images for all experimental groups were generated at the same time, considering the same exposure time and the same contrast/brightness parameters. The images were analyzed blindly with 20× objectives. The analyses for each animal were performed using three non-overlapping separate images of posterior lobules IV/V, VI, and/or VII and were counted manually using Adobe Photoshop software 22.0.0 (Adobe Systems Software Ireland Limited, Dublin, Republic of Ireland) with minimal previous manipulation of contrast. Mean values of cleaved caspase 3 cells, based on 1000 DAPI positive cells, were calculated by averaging the values of all sections of the same animal and used to compare the cell numbers of the treated animals with those of the control animals. The cells counted in the ROI of the control animals were used as 100% values as indicated in the figure legends.

### 4.6. Protein Extraction

Proteins were extracted as previously described [46]. The frozen cerebellar tissue was homogenized in RIPA buffer solution and then centrifuged at 3000× *g* (4 °C) for 10 min. The supernatant was used to determine protein concentrations using the Pierce BCA kit (Pierce/Thermo Scientific, Rockford, IL, USA) after incubation at 37 °C for 30 min followed by spectrophotometry at 562 nm.

### 4.7. Enzyme-Linked Immunosorbent Assays (ELISAs) for TNFα

Tumor necrosis factor α (TNFα) concentration was determined in samples of cerebellar homogenates using the rat TNFα/TNFSF1A kit (R&D Systems GmbH, Wiesbaden-Nordenstadt, Germany) according to the manufacturer’s instructions, as previously described [46]. Plates were read at 450 nm, and TNFα concentrations were calibrated from the standard curve and expressed in picograms per milligram of protein.

### 4.8. Thiobarbituric Acid Reactive Substances (TBARS) Assay

Concentrations of markers of lipid peroxidation thiobarbituric acid reactive substances (TBARS) were determined using the TBARS assay kit (Cayman Chemical, Ann Arbor, MI, USA) according to the manufacturer’s instructions. TBARS concentrations as a measure for lipid peroxidation were calculated from a malondialdehyde (MDA) standard curve and normalized to the amount of total protein, as previously described [46].

### 4.9. Statistical Analyses

Analysis of the data was carried out with GraphPad Prism 8.0 software (GraphPad Software, La Jolla, CA, USA). Data were analyzed using a multivariate repeated measures analysis of variance (ANOVA). Depending on which ANOVA test was used, multiple comparisons of means were carried out using Bonferroni’s post hoc test. GraphPad Prism Software was used for the generation of graphs. Data are presented as box and whisker plots, with the line representing the median while whiskers show the data variability outside the upper and lower quartiles. Differences were considered statistically significant at a *p*-value of <0.05.

## 5. Conclusions

In conclusion, dexmedetomidine effectively counteracted the neurotoxicity induced by hyperoxia in the cerebellum of rats and the associated oxidative damage and neuroinflammation. DEX treatment provided effective protection against oxygen toxicity in the cerebellum. DEX modulated hyperoxia-induced oxidative stress, modulated the oxidative stress response, and reduced the overproduction of the proinflammatory cytokine TNFα. The underlying mechanisms by which DEX exerts these neuroprotective effects are based on the interplay of its radical scavenging properties and downstream anti-inflammatory and anti-apoptotic properties. Further preclinical studies are needed to uncover the additional cellular mechanisms and thus establish the basis for its clinical application.

## Figures and Tables

**Figure 1 ijms-24-07804-f001:**
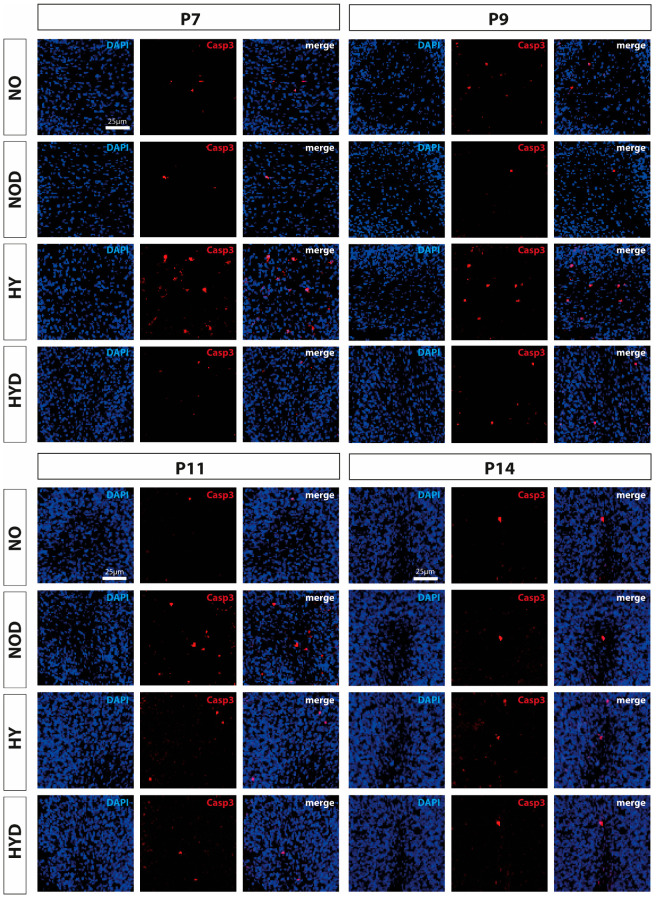
Representative immunofluorescence staining pattern of cerebellar sections with cells staining positive for cleaved caspase 3 (CASP3, red) after 24 h of hyperoxia (80% O_2_) shown for P7 immediately after hyperoxia and for P9, P11, and P14 with survival at normoxic environmental conditions (21% O_2_). High oxygen (HY) resulted in increased cell death on postnatal days 7 and 9 compared with controls under normoxia (NO). Pretreatment with DEX (HYD) reduced cell death induced by high oxygen. DEX under control conditions appeared to increase cell death at P11 (NOD). Nuclear staining was performed with DAPI (blue). Scale bar 25 μm.

**Figure 2 ijms-24-07804-f002:**
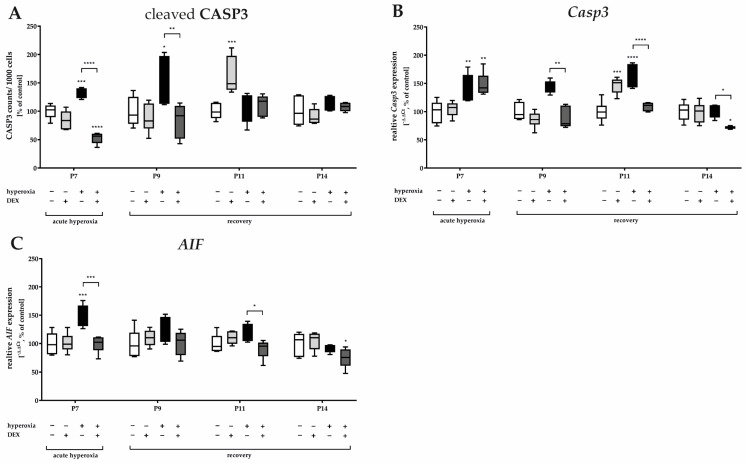
Representation of quantification of (**A**) cleaved caspase 3 (CASP3+) positive cells as a ratio to 1000 DAPI+ cells in cerebellar slices as well as from transcripts of cerebellar homogenates of (**B**) *Casp3* and (**C**) *AIF*, depicted for P7 immediately after hyperoxia (80% O_2_) and for P9, P11, and P14 with survival under normoxic environmental conditions (21% O_2_). Experimental groups are shown for hyperoxia exposure (black bars), hyperoxia with dexmedetomidine (dark gray bars), and dexmedetomidine with normoxia (light gray bars), compared with control animals treated with normoxia and sodium chloride (white bars). Data are normalized to levels in rat pups exposed to normoxia at each time point (control 100%, white bars) and the 100% values are 3.2 (P7), 3.7 (P9), 3.3 (P11), and 1.7 (P14) CASP3+ cells/1000 counted DAPI+ cells per regions of lobules. *n* = 6/group. * *p* < 0.05, ** *p* < 0.01, *** *p* < 0.001, **** *p* < 0.0001 (ANOVA, Bonferroni’s post hoc test).

**Figure 3 ijms-24-07804-f003:**
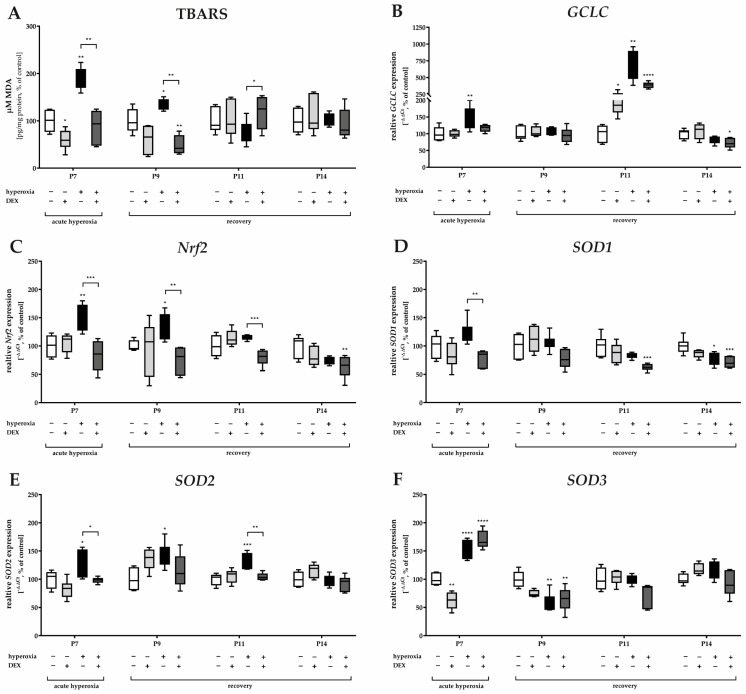
Representation of quantification of (**A**) TBARS in cerebellar protein homogenates and transcripts of cerebellar homogenates of (**B**) *GCLC*, (**C**) *Nrf2*, (**D**) *SOD1*, (**E**) *SOD2*, and (**F**) *SOD3*, depicted for P7 immediately after hyperoxia (80% O_2_) and for P9, P11, and P14 with survival under normoxic environmental conditions (21% O_2_). Experimental groups are shown for hyperoxia exposure (black bars), hyperoxia with dexmedetomidine (dark gray bars), and dexmedetomidine with normoxia (light gray bars), compared with control animals treated with normoxia and sodium chloride (white bars). Data are normalized to levels in rat pups exposed to normoxia at each time point (control 100%, white bars). *n* = 6/group. * *p* < 0.05, ** *p* < 0.01, *** *p* < 0.001, **** *p* < 0.0001 (ANOVA, Bonferroni’s post hoc test).

**Figure 4 ijms-24-07804-f004:**
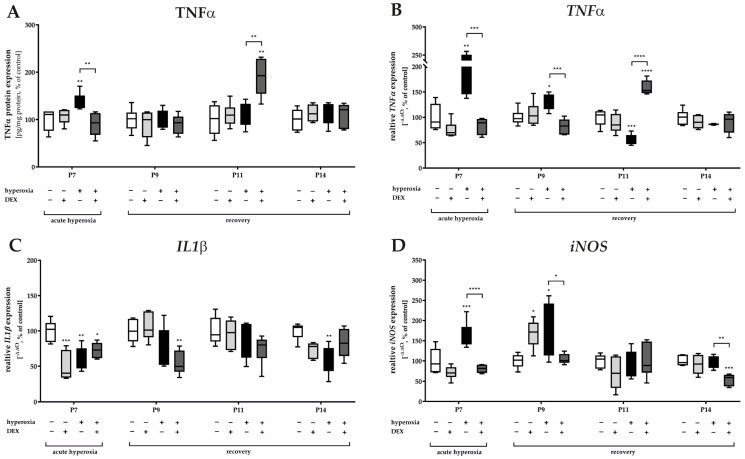
Representation of quantification of (**A**) TNFα in cerebellar protein homogenates and transcripts of cerebellar homogenates of (**B**) *TNFα*, (**C**) *IL1β*, and (**D**) *iNOS* depicted for P7 immediately after hyperoxia (80% O_2_) and for P9, P11, and P14 with survival under normoxic environmental conditions (21% O_2_). Experimental groups are shown for hyperoxia exposure (black bars), hyperoxia with dexmedetomidine (dark gray bars), and dexmedetomidine with normoxia (light gray bars), compared with control animals treated with normoxia and sodium chloride (white bars). Data are normalized to levels in rat pups exposed to normoxia at each time point (control 100%, white bars). *n* = 6/group. * *p* < 0.05, ** *p* < 0.01, *** *p* < 0.001, **** *p* < 0.0001 (ANOVA, Bonferroni’s post hoc test).

**Table 1 ijms-24-07804-t001:** Sequences of oligonucleotides.

	Oligonucleotide Sequence 5′-3′	Accession No.
	*AIF*	
forward	CACAAAGACACTGCAGTTCAGACA	NM_031356.1
reverse	AGGTCCTGAGCAGAGACATAGAAAG	
probe	6-FAM-AGAAGCATCTATTTCCAGCC-TAMRA	
	*Casp3*	
forward	ACAGTGGAACTGACGATGATATGG	NM_012922.2
reverse	AATAGTAACCGGGTGCGGTAGA	
probe	6-FAM-ATGCCAGAAGATACCAGTGG-TAMRA	
	*GCLC*	
forward	GGAGGACAACATGAGGAAACG	NM_012815.2
reverse	GCTCTGGCAGTGTGAATCCA	
probe	6-FAM-TCAGGCTCTTTGCACGATAA-TAMRA	
	*HPRT*	
forward	GGAAAGAACGTCTTGATTGTTGAA	NM_012583.2
reverse	CCAACACTTCGAGAGGTCCTTTT	
probe	6-FAM-CTTTCCTTGGTCAAGCAGTACAGCCCC-TAMRA	
	*IL1* *β*	
forward	CTCCACCTCAATGGACAGAACA	NM_031512.2
reverse	CACAGGGATTTTGTCGTTGCT	
probe	6-FAM-CTCCATGAGCTTTGTACAAG-TAMRA	
	*iNOS*	
forward	AGCTGTAGCACTGCATCAGAAATG	NM_012611.3
reverse	CAGTAATGGCCGACCTGATGT	
probe	6-FAM-CAGACACATACTTTACGCCAC-TAMRA	
	*Nrf2*	
forward	ACTCCCAGGTTGCCCACAT	NM_031789.2
reverse	GCGACTCATGGTCATCTACAAATG	
probe	6-FAM-CTTTGAAGACTGTATGCAGC-TAMRA	
	*SOD1*	
forward	CAGAAGGCAAGCGGTGAAC	NM_017050.1
reverse	CCCCATATTGATGGACATGGA	
probe	6-FAM-TACAGGATTAACTGAAGGCG-TAMRA	
	*SOD2*	
forward	GACCTACGTGAACAATCTGAACGT	NM_017051.2
reverse	AGGCTGAAGAGCAACCTGAGTT	
probe	6-FAM-ACCGAGGAGAAGTACCACGA-TAMRA	
	*SOD3*	
forward	GGAGAGTCCGGTGTCGACTTAG	NM_012880.1
reverse	CTCCATCCAGATCTCCAGGTCTT	
probe	6-FAM-CTGGTTGAGAAGATAGGCGA-TAMRA	
	*TNF* *α*	
forward	CCCCCAATCTGTGTCCTTCTAAC	NM_012675.3
reverse	CGTCTCGTGTGTTTCTGAGCAT	
probe	6-FAM-TAGAAAGGGAATTGTGGCTC-TAMRA	

Abbreviations: apoptosis-inducing factor, mitochondria associated 1 (AIF), caspase 3 (Casp3), glutamate-cysteine ligase, catalytic subunit (GCLC), hypoxanthine-guanine phosphoribosyl-transferase (HPRT), inducible nitric oxide synthase (iNOS), interleukin 1 beta (IL1β), NFE2 like bZIP transcription factor 2 (Nrf2), superoxide dismutase (SOD), tumor necrosis factor (TNFα), 6-carboxyfluorescein (6-FAM), and tetramethylrhodamine (TAMRA).

## Data Availability

The data used to support the findings of this study are available from the corresponding author upon request. The analyzed data used to create the graphs and statistical evaluation are attached in the supplemented material of this work.

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
