# Peer review of "Dexmedetomidine Protects Cerebellar Neurons against Hyperoxia-Induced Oxidative Stress and Apoptosis in the Juvenile Rat"

_ijms, 2023, doi:10.3390/ijms24097804_

Round 1

Reviewer 1 Report

The authors demonstrated the increase in apoptosis in the cerebellum during and after the hyperoxia. This detrimental effect of hyperoxia was to some degree rescued by DEX. The authors also demonstrated the effect on the inflammatory mediators and on the oxidative stress response. Although the rationale of the study was clearly set and experiments were well-performed, some results are confusing. Therefore, the beneficial effect of DEX on oxidative stress is not very conclusive. I look forward to the answers from the authors for the following comments:

1. Fig 1: The density of DAPI staining is not uniform in different panels (for example, P7 NOD v P7 HYD). Is it because the staining region of the cerebellum is different? Additionally, the authors may co-stain Casp3 with additional neuronal/cerebellar neuron marker to improve the uniformity of the staining. It is difficult to say if the apoptotic cells are neurons or glial cells.

2. Fig 1 and 2A: As the images show increased expression of Casp3+ cells in control conditions by DEX (at P11), the authors may include some justification for this anomaly.

3. Line 271: “Consistent with oxidative stress and cell death, the pro-inflammatory cytokine TNFα was induced”. The data, however, shows TNF-a (protein) was increased only at P7 and TNF-a (mRNA) was increased at P7 and P9. Surprisingly, it was decreased at P11 by hyperoxia. Authors may provide justification for this.  

4. Line 302-304: “Data also suggest that the Nrf2/NLRP3 inflammasome signaling pathway is involved. DEX was able to inhibit the expression of NLRP3 by activating Nrf2 and inhibiting proinflammatory mediators”. This statement does not match with the data. NLRP3 positively controls the expression of IL-1β. DEX did not inhibit the expression of IL-1β compared to hyperoxic conditions at any time point. It would be better if the authors also check for the expression of NLRP3.

5. Interestingly, TNF-a and IL-1β levels are exactly opposite to each other which is very confusing. Both of them are pro-inflammatory mediators. Authors may want to explain this phenomenon.

6. The authors have only demonstrated cell death due to apoptosis (Fig 1). However, the overall cell death of neurons or glia cells has not been demonstrated. It is mentioned in the discussion that “oxidative-stress-induced cell death is accompanied by increased levels of proinflammatory cytokines”. It would be good to show if there is non-apoptotic cell death as well.

Reviewer 2 Report

Summary

The manuscript by Robert Puls et al. analyses the effects of dexmedetomidine (DEX) during hyperoxia-induced oxidative stress in the juvenile rats. The authors describe decreased expression and cleavage of Caspase 3, lower oxidative stress and reduced inflammatory response upon DEX treatment. The experiments were performed in Wistar rat pups.

Major comments

1.    Amelioration of hypoxia-induced neurotoxicity, including decreased numbers of caspase-3+ cells and decreased cleavage of caspase-3 by dexmedetomidine treatment have previously been described, e.g. by Wanying Pan et al. (https://doi.org/10.1007/s10571-015-0315-2). The relevant literature has not been cited by the authors.

2.    Reduction of oxidative stress in neonatal rat brais by dexmedetomidine treatment has been described in 2015 by the authors (http://dx.doi.org/10.1155/2015/530371).

3.    Dose-dependent reduction of inflammation by analysis of IL-1β on mRNA and protein level by dexmedetomidine treatment has been described in 2015 by the authors (http://dx.doi.org/10.1155/2015/530371).

Minor comments

4.      Judgmental non-scientific phrases such as “more impressively”, “interesting”, etc. should be avoided in the results section.

The text is linguistically understandable for the most part. Slang, judgmental and unscientific terms should be removed. Correction of some typographical errors is necessary.

Reviewer 3 Report

English is very generally very good.

Round 2

Reviewer 1 Report

The manuscript has been modified according to the comments. I am satisfied with the responses to the comments by the authors. 

Reviewer 2 Report

Thanks for the quick response to the comments. There are no further or new comments on the manuscript.

There are no further or new comments on the quality of english.